# Multivariate analysis of variegated expression in Neurons: A strategy for unbiased localization of gene function to candidate brain regions in larval zebrafish

**Hannah Shoenhard**[ORCID]**, Michael Granato**[ORCID]*

Department of Cell and Developmental Biology, Perelman School of Medicine, University of Pennsylvania, Philadelphia, Pennsylvania, United States of America

* granatom@pennmedicine.upenn.edu

## Abstract

Behavioral screens in model organisms have greatly facilitated the identification of genes and genetic pathways that regulate defined behaviors. Identifying the neural circuitry via which specific genes function to modify behavior remains a significant challenge in the field. Tissue- and cell type-specific knockout, knockdown, and rescue experiments serve this purpose, yet in zebrafish screening through dozens of candidate cell-type-specific and brain-region specific driver lines for their ability to rescue a mutant phenotype remains a bottleneck. Here we report on an alternative strategy that takes advantage of the variegation often present in Gal4-driven UAS lines to express a rescue construct in a neuronal tissue-specific and variegated manner. We developed and validated a computational pipeline that identifies specific brain regions where expression levels of the variegated rescue construct correlate with rescue of a mutant phenotype, indicating that gene expression levels in these regions may causally influence behavior. We termed this unbiased correlative approach Multivariate Analysis of Variegated Expression in Neurons (MAVEN). The MAVEN strategy advances the user's capacity to quickly identify candidate brain regions where gene function may be relevant to a behavioral phenotype. This allows the user to skip or greatly reduce screening for rescue and proceed to experimental validation of candidate brain regions via genetically targeted approaches. MAVEN thus facilitates identification of brain regions in which specific genes function to regulate larval zebrafish behavior.

## Introduction

Behaviors are mediated by ensembles of interconnected neurons and glia that make up neuronal circuits. A genetic mutation may disrupt the development and/or function of one or more specific circuit elements, thereby producing a behavioral phenotype. From a basic research perspective, identifying circuit elements that mediate behavioral phenotypes aids in understanding the interplay between genetic mutations, cellular functioning, circuit activity, and

**Data Availability Statement:** All relevant data are within the paper and its Supporting information files. Specifically, data used for figure creation are in S2, sample data used for multivariate analysis

and all multivariate analysis code and associated files are available in a zipped folder in the Supporting information files.

**Funding:** This research was supported by National Institutes of Health (NIH) grants R01NS118921, R01MH109498, and R21MH103545 (to M.G.) and training grants T32MH017168-35 and T32GM007517-37 (to H.S.). The funders had no role in study design, data collection and analysis, decision to publish, or preparation of the manuscript.

**Competing interests:** The authors have declared that no competing interests exist.

behavior [1]. From a translational perspective, therapies must be targeted to the correct cells in order to succeed, so identifying phenotypically relevant cell types and brain regions in model organisms is critical [2]. However, this critical task remains a major challenge in a major vertebrate model organism, the larval zebrafish. Here, we a present strategy in larval zebrafish for identifying the brain region(s) via which a specific gene regulates a behavioral phenotype.

In genetic model organisms various specifically targeted approaches, such as genetic knockout, knockdown, and rescue, are used to establish the cell type(s) and brain region(s) in which specific genes function to regulate behavior. In rescue approach, a promotor or driver is used to induce expression of a rescue construct containing the target gene in a specific population of cells in the context of an animal that is otherwise mutant for the target gene. If this manipulation normalizes, or "rescues" the typical mutant phenotype, the gene's function has been successfully localized to that specific population of cells. Indeed, in larval zebrafish hundreds of Gal4 drivers are available for inducing gene expression in specific defined populations, making it a seemingly feasible option to screen through available Gal4s until rescue is achieved and therefore the relevant cells are identified [3–8]. However, two major hurdles stand in the way of rescue screens in larval zebrafish.

First, rescue screens are logistically challenging in zebrafish. Cell-type-specific rescue screens must take place on a mutant genetic background, meaning that each individual Gal4 line must be crossed into the mutant background, raised to adulthood over 3 or more months, genotyped, and finally crossed to a UAS rescue construct carrier also on the mutant background. Negative results in a rescue experiment do not definitively rule out the targeted cellular population because the possibility always remains that the target gene was not expressed at sufficient levels to rescue the phenotype. The number of Gal4 lines to be tested may also present a challenge. In *Drosophila* the number of Gal4 lines screened for rescue of a mutant phenotype can exceed several dozen [9]. In zebrafish, an unbiased cell-type-specific rescue screen of more than a few dozen Gal4 lines would be prohibitively costly for most labs in terms of time, labor, and tank space. For these reasons, few if any genetic rescue screens have been used to identify the specific brain regions in which behaviorally-relevant genes function in larval zebrafish.

A second hindrance to Gal4 x UAS-based rescue screens in zebrafish is the epigenetic silencing of UAS lines [10–12]. Variegated expression of transgenic lines in zebrafish and silencing throughout generations has been observed for as long as transgenic lines have been made [13, 14], but UAS lines are possibly particularly susceptible due to methylation of their repetitive promoter element. Some UAS lines continue to be expressed in all targeted cells across multiple generations, while others are rapidly silenced, while others express in a variegated fashion, producing a different pattern in every larva [15, 16]. When a UAS construct is employed for rescue of a mutant phenotype, variegation can produce variability, undermining confidence in results. Historically variegation has been regarded as a disadvantage of the Gal4 x UAS system in zebrafish and has motivated the development of alternative combinatorial expression systems [17]. Our strategy utilizes variegated and/or mosaic expression throughout the brain for a computational analysis, and thus turns the natural variegation of some UAS constructs into an advantage.

Some conditions must be met before our strategy can be successfully utilized. First, we assume that the broad tissue or cell type in which the target gene functions in behavior—e.g. neurons, oligodendrocytes, astrocytes—has already been identified via cell-type-specific rescue, and that the user wishes to find the specific brain region(s) where the gene exerts its effects. Second, we assume that the target gene can be expressed in a variegated and/or mosaic fashion in the cell type of interest, either with a variegated UAS construct, via DNA or RNA microinjection, or another technique. Third, we assume that the ectopically-expressed target

gene can be visualized within the context of the brain, either by directly tagging the gene with a fluorophore or epitope or by driving expression of a fluorophore from the same genetic construct. Once these assumptions are met, our strategy can be applied.

We call our strategy Multivariate Analysis of Variegated Expression in Neurons (MAVEN). MAVEN can be used when variegated, cell-type-specific expression of a rescue construct leads to variable rescue of a behavioral phenotype (Fig 1A). Larvae are collected in two behavioral groups: unrescued and rescued (Fig 1B). We assume that the rescued larvae express the rescue construct in the functionally relevant population of cells, while the unrescued larvae do not. Next, we image the expression pattern of the variegated rescue construct in each group of

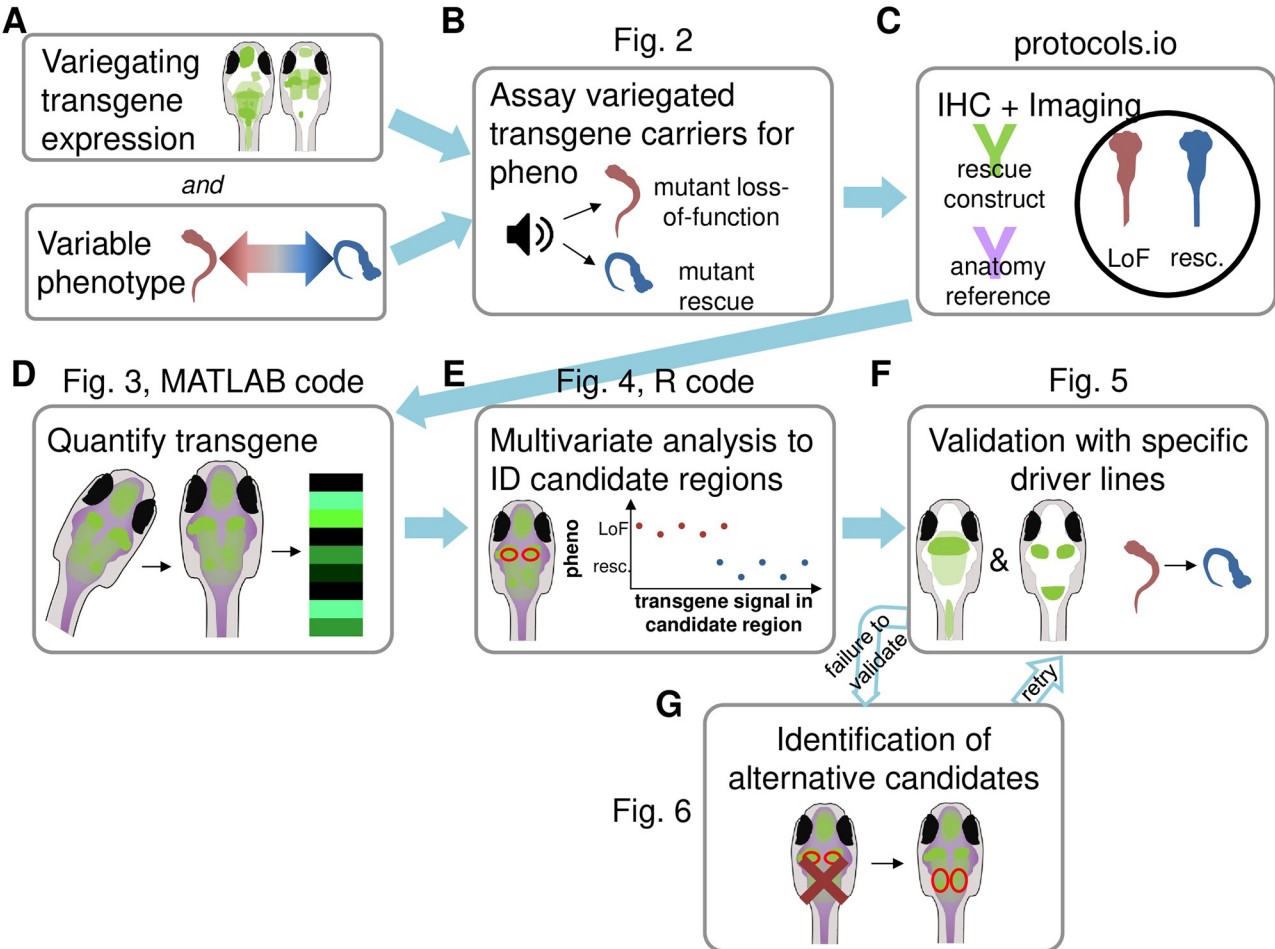

**Fig 1. Overview of experimental workflow.** A) Before MAVEN can be performed, a variegated, labeled expression construct for the gene of interest that has variable effects on a phenotype of interest is required. B) The first step of MAVEN is to collect two groups of larvae, each expressing the variegated transgene, but displaying different behavioral phenotypes. In this example, some mutant larvae fail to express the transgene in the relevant cells, and therefore display the typical mutant phenotype (red), while other mutant larvae express the transgene in relevant cells and therefore display a WT-like, rescued behavioral phenotype (blue). C) After larvae are collected in each phenotype group, their tails are trimmed distinctively so they can be identified after immunohistochemical staining in a single tube to avoid batch artifacts. Antibodies for the rescue construct (in this example, GFP) and an anatomical reference stain are applied. D) Next, the brains are aligned to a 3D brain atlas using the reference stain and the rescue construct signal in identified brain regions is quantified. We give an example with our own data in Fig 3. We provide MATLAB code for this stage of the protocol. E) Multivariate analysis is performed to identify specific brain regions in which GFP signal levels correlate with larval phenotype. These are candidate regions which may mediate the function of the gene of interest. We show example results of our analysis in Fig 4, and provide R code for this stage. F) Validation is performed by identifying more specific Gal4 drivers for candidate regions. If the phenotype of interest can be rescued by driving expression solely in these regions, then a region that mediates the gene's effects on phenotype has been successfully identified. We show how we validated our results in Fig 5. G) If a specific region fails to validate, we offer methods to identify alternative candidate regions in Fig 6.

brains (Fig 1C). Brains are aligned to a standard reference atlas to allow quantification of rescue construct expression in defined brain regions (Fig 1D). Multivariate analysis then produces a short list of specific brain regions where rescue construct expression differs between unrescued and rescued larvae (Fig 1E). We reason that in these brain regions, gene expression may causally influence behavior, making them good candidates for more specific rescue experiments. Specific rescue experiments validate the causal role for MAVEN-identified candidate brain regions. To perform these experiments, using the brain atlas, we identify Gal4 lines that drive expression in candidate regions. These Gal4 lines are then used for region-specific rescue experiments to test whether gene activity in these regions is causally relevant for behavior (Fig 1F). Thus, even without highly laborious Gal4 x UAS screens, MAVEN facilitates localization of genetic function to specific brain region(s).

In the example we present in this protocol, our analysis produced a single candidate brain region. We identified two relatively specific Gal4 lines that we used to validate this candidate region's role in behavior as a site of action for our gene of interest. However, we recognize that validation may not succeed with every candidate region. We therefore supply possible methods for identifying alternative candidate regions, should the first candidates tested fail to validate (Fig 1G). We also recognize that a variegating UAS rescue construct—which we identified serendipitously for our example gene—may not be available to every potential user of this protocol. We offer some suggestions for how variegating and/or mosaic expression can be deliberately achieved in the "Future improvement of MAVEN" section.

## Materials and methods

All animal experiments were approved by the University of Pennsylvania Institutional Animal Care and Use Committee (IACUC). The protocol described in this peer-reviewed article is published on protocols.io, DOI: dx.doi.org/10.17504/protocols.io.14egn78jyv5d/v1 and is included for printing as S1 File with this article.

## Expected results

### Step 1: Collection of variegated transgene-expressing larvae with divergent phenotypes

This protocol requires comparison of transgene expression patterns between two groups of larvae. Two broad strategies—one based on the rescue of a loss-of-function phenotype and one based on an overexpression phenotype—are available to collect larval populations for comparison. In our case, both the loss-of-function rescue and overexpression strategies were viable options. We assume most users will employ the loss-of-function rescue strategy, as it will likely be available for a larger set of genes than the overexpression strategy, which requires an overexpression phenotype. We therefore focus on the loss-of-function strategy in this manuscript. For those who wish to use a strategy based on an overexpression phenotype, as we did for our example gene, we provide more detailed information and caveats in Step 2 on protocols.io.

Our example target gene, *CaSR*, is necessary for a larval zebrafish to display typical sensorimotor decision-making bias in response to sudden acoustic stimuli [18]. At low stimulus intensities, WT larvae tend to respond to acoustic stimuli with a reorientation behavior, while at high stimulus intensities, they perform an escape behavior (Fig 2A) [18]. *CaSR* mutants, while capable of escapes, always bias towards the reorientation (Fig 2A). Variegated CaSR-EGFP expression in neurons, driven using the Gal4 x UAS system, is sufficient to rescue the reorientation bias of some, but not all, *CaSR* mutants (Fig 2B) [19]. Specifically, we generated larvae by crossing the *αtubulin:Gal4* stable transgenic line [20] to the *14xUAS:CaSR-EGFP*

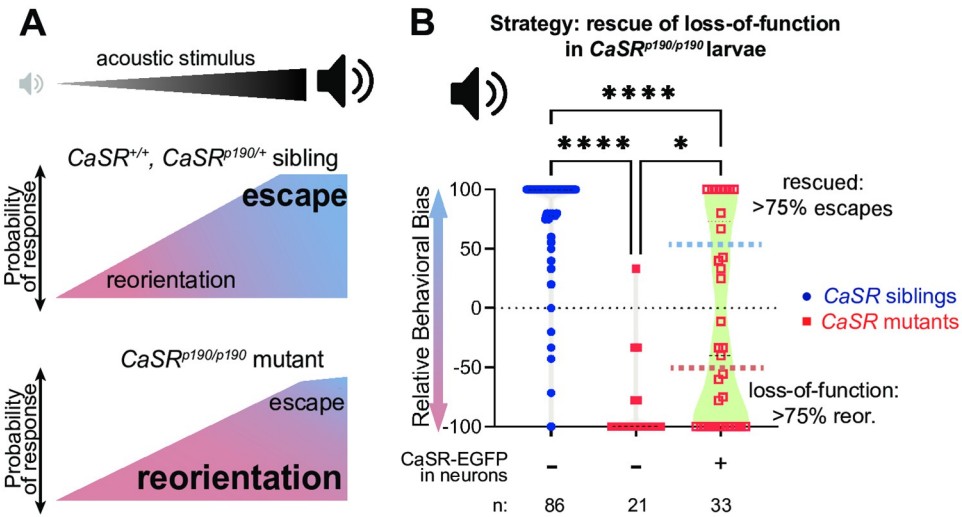

**Fig 2. An example larval collection strategy based on rescue of a loss-of-function mutant phenotype.** A) Schematic of behavioral bias for the escape versus reorientation decision of various larvae. At low stimulus intensities, WT larvae perform predominantly reorientations (pink). As stimulus intensity increases, WT larvae shift their preference towards escapes (light blue). *CaSR* mutant larvae, by contrast, are highly reorientation-biased across all stimulus intensities, although they do occasionally perform escapes. B) Example of a loss-of-function rescue strategy for collecting groups of larvae with divergent phenotypes. When responding to a strong acoustic stimulus, *CaSR* homozygous mutant larvae (red squares) are reorientation-biased relative to sibling controls (blue circles). Overexpressing CaSR in a variegated fashion in neurons in CaSR mutants (empty red squares) sometimes, but not always, rescues the mutant phenotype. We collected larvae in rescued (above light blue line) and non-rescued (below pink line) groups. Some data from Panel B also appears in Shoenhard, Jain, and Granato [19].

stable transgenic line [19]. For the loss-of-function strategy, we divided these *CaSR* mutant larvae into two groups: those that responded to high intensity stimuli with strong bias towards reorientations (loss-of-function phenotype), and those that responded to the same stimuli with strong bias towards escapes (rescue of loss-of-function phenotype) (Fig 2B). For the over-expression strategy, we divided *CaSR* genotypically wild-type larvae that overexpressed *CaSR* in variegated populations of neurons into two groups: those that responded to low-intensity stimuli with bias towards reorientations (typical phenotype) and those that responded to low-intensity stimuli with bias towards escapes (overexpression phenotype). This general approach can be adapted to a wide variety of behaviors other than sensorimotor decision-making, so long as there is some phenotypic variability in mutants expressing the variegated rescue construct.

## Step 2: Quantification of transgene in anatomically-identified brain regions

After collecting larvae in two phenotypic groups, we compare the expression pattern of the rescue construct between each group. To do this, we immunostained larvae for total ERK (for ZBrain atlas registration—see protocols.io Steps 3–12), imaged total ERK and CaSR-EGFP expression pattern on a confocal microscope (see protocols.io Step 13), and then registered them to the ZBrain 3D anatomical atlas (see protocols.io Steps 15–16 and Randlett et al. 2015). We then quantified the CaSR-EGFP signal of all registered larvae across the brain regions in the ZBrain atlas (Fig 3A) using the custom Matlab function QuantifySignalMultipleBrains (available for download in protocols.io Step 17). Next, we imported the data to R for further analysis (example code available for download in protocols.io Step 18).

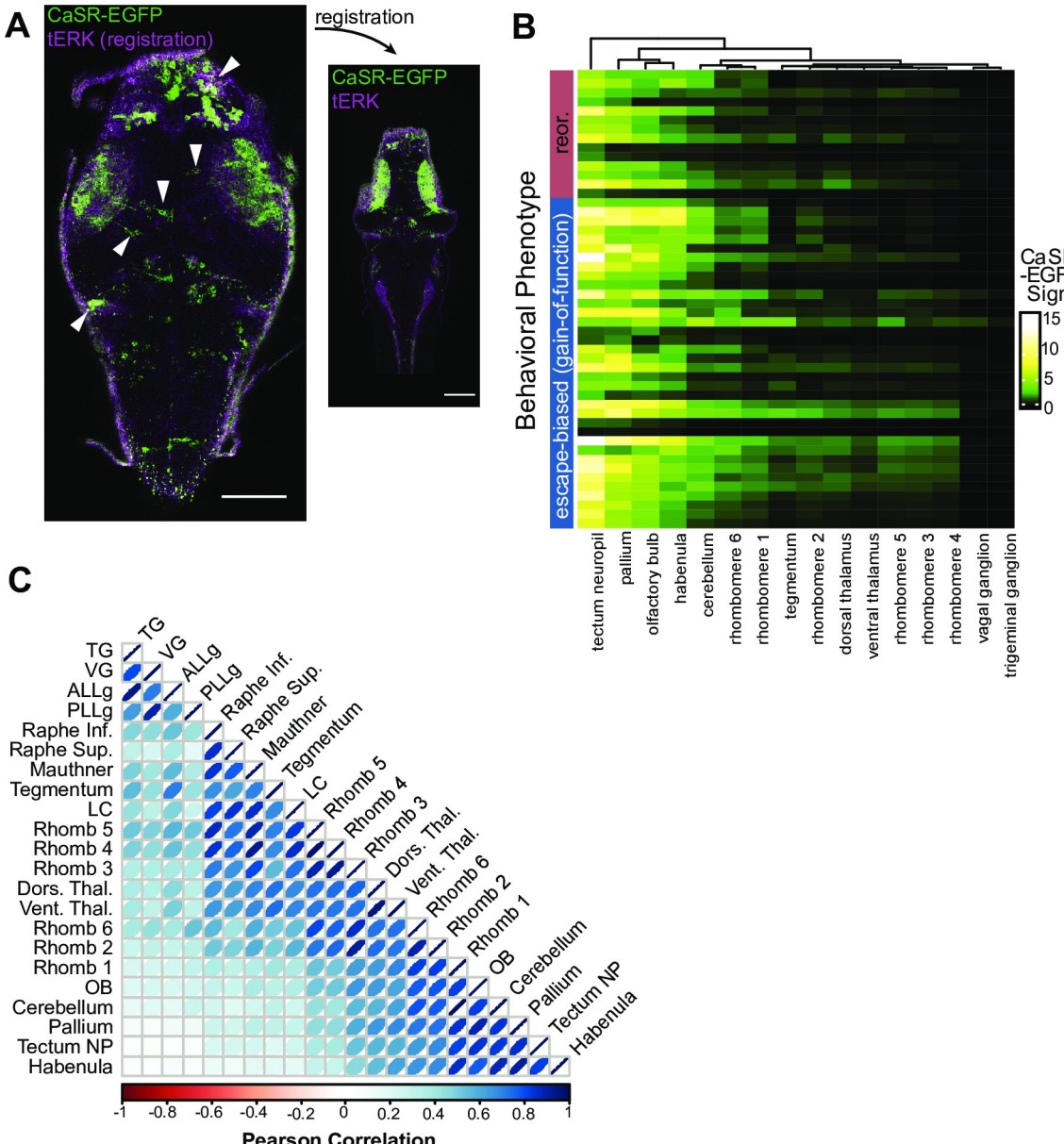

**Fig 3. Patterns of variegation of rescue construct expression in neurons of d5 larvae.** A) CaSR-EGFP (green) and total ERK anatomy reference (magenta) signal in single slices in the dorsal portion of the brain of an *αtubulin*:*Gal4; UAS:CaSR-EGFP* larva. Left and right images are from the same brain: raw image is on the left and brain atlas-registered image is on the right. Note the variegation in CaSR-EGFP expression, with left-right asymmetries highlighted using white arrowheads. Scale bars are 100 um. Due to the deformations that occur during brain atlas registration, the raw image and the registered image do not depict exactly the same anatomical regions, and therefore the exact same cells cannot be identified from image to image. B) Quantification of CaSR-EGFP signal in the brains of n = 50 *αtubulin*:*Gal4> UAS:CaSR-EGFP; CaSR*^+/+ larvae sorted by behavioral phenotype. Signal in each brain region is represented as a gradient ascending through the colors black, green, yellow, and white. C) Pearson correlations of a set of example brain regions calculated on the full dataset of n = 150 larvae. Width and color of ellipses are proportional to the degree of correlation. Regions are ordered by hierarchical clustering using the Ward method. Red = negatively correlated (no brain regions in this group were negatively correlated), white = no correlation, blue = positively correlated.

We rely on the natural variability in expression of our rescue transgene—both between larvae, and between brain regions in single larvae—for our analysis. Some brain regions exhibited high variability in expression between larvae, while a few brain regions nearly always highly expressed our transgene and a few nearly never expressed our transgene (Fig 3B). Likewise, some larvae expressed our rescue transgene highly across almost the whole brain, some barely expressed it at all, and most expressed it variably in different regions. In summary, a high degree of variability existed in the expression of our rescue construct both within and between brains.

For MAVEN, expression levels in each brain region serve as the independent variables, and phenotype is the dependent variable. Multicollinearity, or correlation between independent variables, is a challenge for multivariate analysis because it renders it difficult to discern which of two highly correlated variables (if any) actually predicts the dependent variable. We therefore determined the correlation of CaSR-EGFP signal across a subset of brain regions (Fig 3C). We hypothesized that signal would not be perfectly independent; rather, some larvae would likely have overall more UAS inactivation and some would have less, leading signal in different brain regions to be correlated. We performed pairwise Pearson correlations between a subset of brain regions and found that correlation varied from nearly 0 to nearly 1, with a full range in between (Fig 3C). Few, if any, pairs of brain regions were negatively correlated. Correlation between independent variables informed our selection of a method for multivariate analysis and our interpretation of the multivariate analysis results we obtained.

Our example dataset consists of 172 brains of larvae with various *CaSR* genotypes. Signal within various brain regions exhibits varying degrees of positive correlation, with pairwise Pearson correlations ranging from 0 to 1. For those seeking to apply the MAVEN strategy, each variegated or mosaic construct is likely to display different patterns of correlation. Less correlation between brain regions will provide greater power in subsequent steps of multivariate analysis to determine which brain regions best predict behavioral phenotype. Extremely high levels of correlation could render multivariate analysis uninformative, because no single brain region will be able to predict phenotype better than any other. Therefore, it is critical to examine and consider patterns of variegation before attempting MAVEN and when interpreting results. We achieved success using an n of 50 wild-type larvae for our multivariate analysis (note that users who follow the loss-of-function rescue strategy will employ mutant larvae for their multivariate analyses). Depending on their own datasets' levels of multicollinearity, users may need to adjust their experimental ns accordingly.

## Step 3: Multivariate analysis with LASSO regression

We selected a multivariate analysis method suited to our dataset and our goals. Our goal was to produce a short list of defined candidate brain regions which we could easily validate via more specific Gal4 x UAS expression experiments. We therefore avoided neural network and random forest models, which can produce highly accurate classifiers for datasets, but do not lend themselves to easy interpretation [21, 22]. We also chose not to use dimensionality reduction techniques before analyzing our data. This is because dimensionality reduction techniques, such as PCA, flatten many variables that are easily interpreted (e.g. signal from individual brain regions) into single variables that are more difficult to interpret (e.g. one variable consisting of a weighted amalgamation of signal from many brain regions). Finally, we knew that multicollinearity was present in our dataset (Fig 3B), and wished to use a method suitable for such situations [23]. We therefore selected for our analysis logistic LASSO (least absolute shrinkage and selection operator) regression. LASSO regression, like other forms of multivariate regression, constructs a mathematical model that predicts a dependent variable—

in this case, phenotype—by summing independent variables—in this case, rescue construct expression level in a particular brain region—that are weighted by specific fitted coefficients [24, 25]. Distinctively, LASSO regression reduces to zero coefficients of independent variables that are not sufficiently informative, performing variable selection without first reducing the dimensionality of the dataset. LASSO regression thereby produces an interpretable but sparse model that tolerates a moderate degree of multicollinearity and resists overfitting [23, 25].

For those taking the loss-of-function rescue approach, LASSO regression should be used to compare rescue construct expression levels in brain regions of phenotypically non-rescued versus rescued mutant larvae. We took the overexpression approach, so we used LASSO regression to compare the CaSR expression levels in brain regions of larvae with the overexpression phenotype versus larvae with the typical phenotype. The analysis steps involved in the loss-of-function rescue and overexpression approaches are the same, but the phenotypic groups collected and compared are different. For our overexpression-based analysis, we quantified the variegated neuronal CaSR-EGFP expression levels in n = 251 brain regions of n = 50 wild-type larvae that responded to low-intensity stimuli with either reorientation-biased (typical phenotype) or escape-shifted (overexpression phenotype) behavior (see protocols.io Step 18). We then performed LASSO regression to predict behavioral phenotype using the brain region expression data. The resulting model included a single coefficient for the ZBrain region "Rhombencephalon QRFP Cluster–Sparse", which we refer to as the "Dorsal Cluster–Rhombomere 6" (DCR6). This indicated that CaSR-EGFP expression levels in the DCR6 could help predict a wild-type larva's shift in bias towards escapes. To confirm this assessment, we performed a single two-way ANOVA to assess if "escape-shifted" larvae had more CaSR-EGFP in their DCR6 than "not escape-shifted" larvae across all genotypes. We found a highly significant association between signal in the DCR6 and escape bias overall (main effect of phenotype p<0.0001, with significant within-genotype effects after controlling for multiple comparisons in $CaSR^{+/+}$ and $CaSR^{p190/p190}$ groups) (Fig 4A).

While our initial analysis highlighted the DCR6 as a candidate mediator of the CaSR overexpression phenotype, it was not clear if this brain region might also play a role in rescue of the loss-of-function phenotype. To examine this question, we collected n = 12 $CaSR$ mutant larvae expressing variegated CaSR-EGFP under control of the $\alpha Tubulin$:$Gal4$ driver and sorted them by their responses to strong stimuli: the typical $CaSR$ mutant reorientation bias or the phenotypically rescued escape bias. Indeed, we found a significant association between CaSR-EGFP signal in the DCR6 and phenotypic rescue in $CaSR$ mutants (p = 0.0260, Mann-Whitney U test, Fig 4B).

As a negative control, we performed the same Two-way ANOVA analyses on a different brain region that was not implicated by LASSO regression. For this we selected the locus coeruleus (LC) because it also resides in the hindbrain and is roughly the same size as the DCR6. No significant association was observed between behavioral phenotype and CaSR-EGFP signal in the LC (main effect of phenotype p = 0.0738, no significant effects of phenotype within any genotype) (Fig 4C). There was also no significant correlation between signal in the LC and rescue of decision-making in $CaSR$ mutants (p = 0.0649, Mann-Whitney U test, Fig 4D). Both of these comparisons approached, but did not reach, statistical significance. We theorize that this is due to the multicollinearity in the dataset, meaning there is a moderate degree of correlation between CaSR-EGFP levels in the LC and the DCR6 across all larvae (Pearson correlation R-squared = 0.233, p = 0.0055). This near-significant result underlines the importance of choosing an unbiased method, like LASSO regression, to identify the most promising brain regions for further analysis. A hypothesis-driven approach might have selected the LC as a brain region of interest, given the potential role for the neuromodulator norepinephrine in a decision-

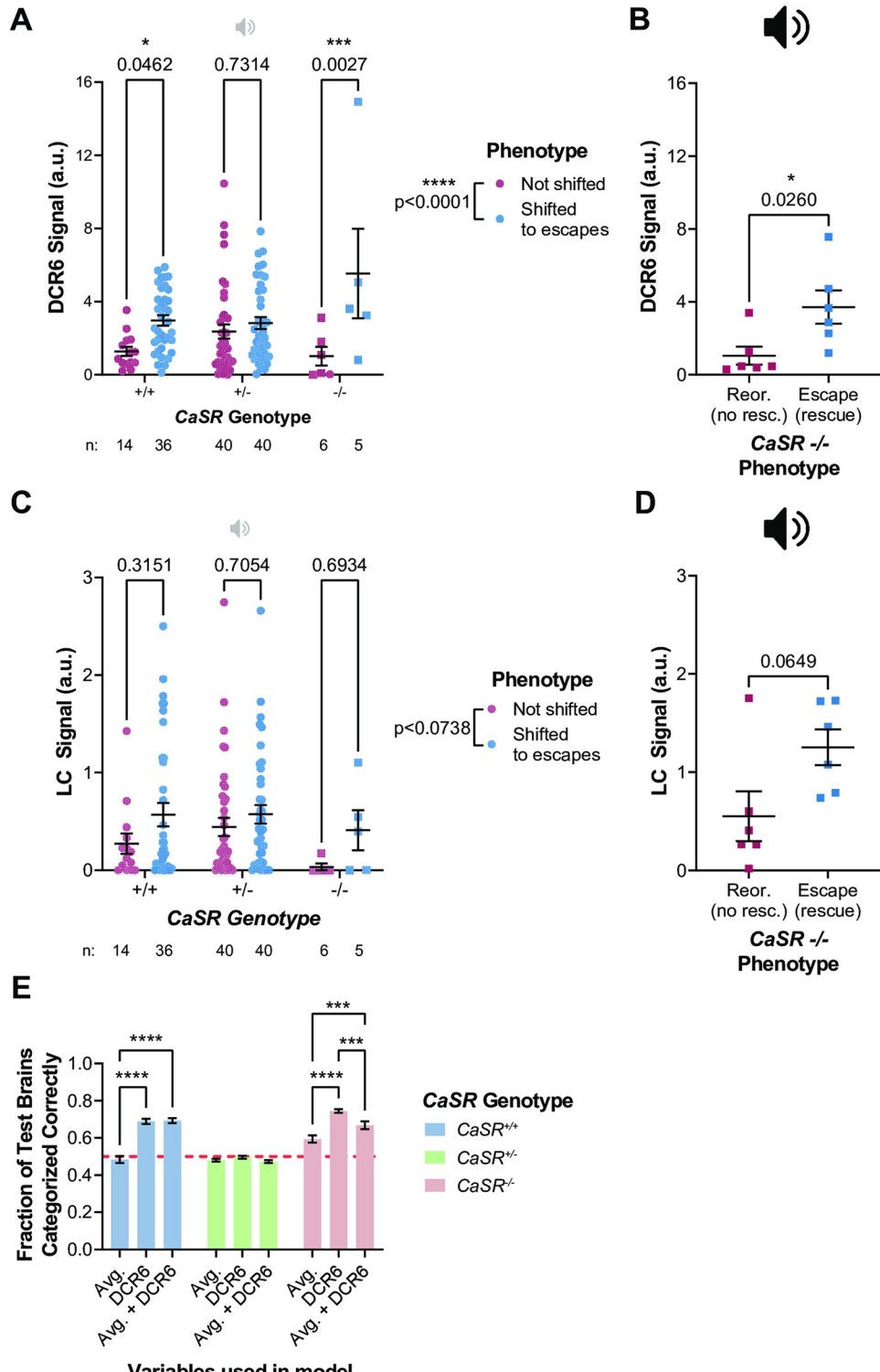

**Fig 4. Identification of the Dorsal Cluster—Rhombomere 6 as a candidate site for CaSR-dependent decision-making.** A) Normalized fluorescence intensity signal in the DRC6 of larvae of various *CaSR* genotypes that either displayed the typical phenotype (red) or were escape-shifted (phenotype caused by CaSR overexpression) (blue) in response to a low-intensity, primarily reorientation-evoking stimulus. Presence of the overexpression phenotype significantly (p<0.0001) predicts DCR6 signal by Two-way ANOVA. B) Normalized fluorescence intensity signal in the Dorsal Cluster–Rhombomere 6 of *CaSR* mutants whose behavioral phenotype was unrescued (red) vs rescued

(blue), e.g. performed predominantly escapes in response to a strong, escape-evoking stimulus. Rescue significantly predicts DCR6 signal by Mann-Whitney U test. C) Normalized fluorescence intensity signal in the locus coeruleus (LC) of larvae of various *CaSR* genotypes that either displayed the typical phenotype (red) or were escape-shifted (overexpression phenotype) (blue) in response to a low-intensity stimulus. Presence of the overexpression phenotype does not significantly predict (p = 0.0738) LC signal by Two-way ANOVA. D) Normalized fluorescence intensity signal in the locus coeruleus of *CaSR* mutants whose behavioral phenotype was unrescued (red) vs. rescued (blue), e.g. performed predominantly escapes in response to a strong, escape-evoking stimulus. Rescue does not significantly predict LC signal by Mann-Whitney U test (p = 0.0649). E) Accuracy of logistic regression models used to predict phenotype of $CaSR^{+/+}$ (blue), $CaSR^{p190/+}$ (green), or $CaSR^{p190/p190}$ (red) larvae using the average CaSR-EGFP signal across the whole brain, the signal only in the DCR6, or both. $CaSR^{+/+}$ and $CaSR^{p190/p190}$ phenotypes could be more accurately predicted with the DCR6 than with the average signal in the whole brain, and using both variables in the model did not improve accuracy compared to the model that used only the DCR6. $CaSR^{p190/+}$ phenotypes were not accurately predicted by any of the three models. Data in panels A and B also appears in Shoenhard, Jain, and Granato [19].

making task. However, unbiased analysis reveals that other regions, like the DCR6, are most strongly linked.

Because signal in various brain regions tends to be positively correlated, we also considered the possibility that overall CaSR-EGFP signal in the brain could predict phenotype. If this were the case, then any brain region that served as a good proxy for average whole-brain expression might be selected by LASSO regression. To evaluate this possibility, we averaged the signal across all brain regions in a single larva and performed bootstrapped univariate logistic regression over 100 randomly drawn training subsets of our data to predict phenotype. We found this univariate average-signal model performed no better than chance on test sets of $CaSR^{+/+}$ or $CaSR^{p190/+}$ larvae (bootstrapped analysis accuracy on test larvae = 48% +/- 1.9% for $CaSR^{+/+}$ larvae and 48% +/- 0.8% for $CaSR^{p190/+}$, mean +/- SEM), but performed better than chance in $CaSR^{p190/p190}$ larvae (bootstrapped analysis accuracy = 59% +/- 1.9%). By contrast, when we performed the same analysis using only signal in the DCR6 as the independent variable, we found that the phenotype of $CaSR^{+/+}$ larvae was predicted with 69% +/- 1.5% accuracy and $CaSR^{p190/p190}$ larvae were predicted with 74% +/- 1.0% accuracy, consistent with our earlier Two-way ANOVA results. Again consistent with the Two-way ANOVA, DCR6 signal was not informative for $CaSR^{p190/+}$ larvae, with an accuracy of 50% +/- 0.8%, no better than chance. Finally, we considered if average CaSR-EGFP signal in the brain, while not informative by itself, could help improve the accuracy of a DCR6 signal model. We therefore bootstrapped models including both average signal and DCR6 signal. However, we found that this model did not improve upon the DCR6 alone model in any genotype (Fig 4E). The MAVEN strategy has thus identified a hindbrain region, the DCR6, where CaSR-EGFP expression levels are highly correlated with escape-shifted decision-making, independent of the average signal across all brain regions.

## Step 4: Candidate brain region validation

While the MAVEN strategy is capable of identifying brain regions where a genetic rescue construct expression strongly correlates with phenotype, it does not necessarily imply that these regions are causally involved with behavior. To test whether target gene expression in a specific brain region is truly causatively influencing behavior, the target gene should be specifically expressed in that brain region and its phenotypic effects assayed. Since the ZBrain atlas is used in MAVEN, this tool can be searched to identify Gal4 drivers that express in the candidate region(s) of interest (https://zebrafishatlas.zib.de/lm). When completely specific Gal4 lines for a given brain region are not available, testing rescue with multiple Gal4s that express in a common target region and different off-target regions can potentially be informative. Additionally,

since functional heterogeneity is characteristic of many brain regions, multiple Gal4s that seemingly target the same region may in fact express in different molecular subpopulations. Therefore, testing multiple Gal4s that seem to express in the same region can potentially provide additional information about which cell type(s) and specific circuit elements are relevant for genetic function. In our case, we identified two Gal4 drivers that expressed in the DCR6 and some immediately bordering regions (Fig 5A) and used them to validate the causative role of CaSR in this region.

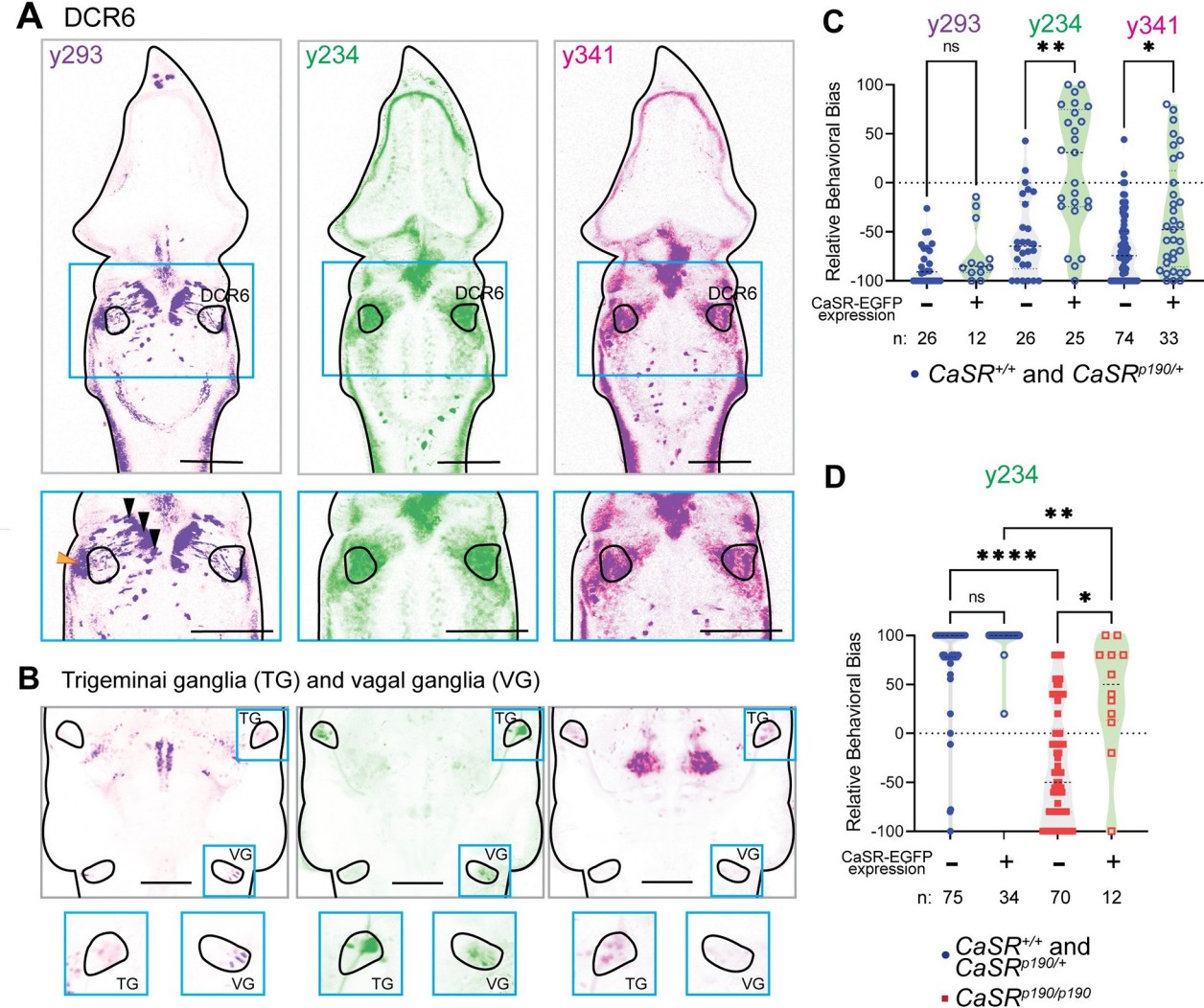

**Fig 5. Example validation of a single Gal4-defined brain region as a mediator of a genetic behavioral phenotype.** A) Expression patterns of the y293 (purple), y234 (green), and y341 (magenta) Gal4s relative to the DCR6 in Slice 122 of ZBrain 2.0 (https://zebrafishatlas.zib.de/lm) [8, 26]. Scale bars are 100 um. Below, enlarged images of the DCR6. The y293 Gal4 labels apparent cell bodies of a population medial to the DCR6 (black arrowheads) that project through the DCR6 and terminate in a region lateral to the DCR6 (orange arrowheads). However, cell bodies within the DCR6 are labeled only sparsely. By contrast, the y234 and y341 Gal4s both label cell bodies within the DCR6 and immediately surrounding it, y234 to a slightly greater extent than y341. B) Maximum projections from ZBrain slices 12–64 (ventral portion of the brain) in the y293 (blue), y234 (green), and y341 (magenta) Gal4s. All three Gal4s drive expression in the trigeminal ganglion (TG), while only y293 and y234 drive expression in the vagal ganglion (VG). Scale bar = 100 um. C) CaSR overexpression escape-shifted phenotype is exhibited by GFP+ *y234; UAS:CaSR-EGFP*; *CaSR* sibling (*CaSR^{+/+}* and *CaSR^{p190/+}*) larvae responding to a low-intensity acoustic stimulus [19]. D) Partial rescue of CaSR loss-of-function phenotype in sorted GFP+ *y234; UAS:CaSR-EGFP*; *CaSR^{p190/p190}* larvae responding to a strong acoustic stimulus. *CaSR* sibling (*CaSR^{+/+}* and *CaSR^{p190/+}*) larvae in blue; *CaSR^{p190/p190}* mutant larvae in red. Data from Panel D and some data from Panel C also appear in Shoenhard, Jain, and Granato [19].

We chose to validate the role of the DCR6 in decision-making by driving *UAS:CaS-R-EGFP* with Gal4s that target this region. If CaSR expression driven by these Gal4s could drive the overexpression phenotype in WT larvae and/or rescue the loss-of-function phenotype in *CaSR* mutant larvae, a role for CaSR in this Gal4-defined region in decision-making would be confirmed. We manually searched the ZBrain atlas [8, 26] for Gal4s that drove expression in the DCR6. While completely specific Gal4s (e.g. those that drive expression across the entire DCR6, but not in any other brain regions) were not available, we obtained two lines, *y234* and *y341*, with expression patterns that included the DCR6 (Fig 5A). These lines also both drove expression in the trigeminal and/or vagal ganglia (Fig 5B), so we obtained a negative control line, *y293*, that drove expression in both ganglia (Fig 5B), but drove only sparse expression in and around the DCR6 (Fig 5A). We then tested whether driving CaSR in populations defined by these Gal4s was sufficient to evoke the *CaSR* overexpression phenotype in siblings and/or rescue the *CaSR* loss-of-function phenotype in mutants.

Driving CaSR with either DCR6-targeting Gal4 line was sufficient to shift decision-making of *CaSR* sibling larvae towards escapes (Fig 5C). By contrast, driving CaSR within the trigeminal and vagal ganglia without the DCR6 failed to shift decision-making (Fig 5C). Furthermore, we demonstrated that CaSR expression driven by the DCR6-targeting *y234* line was sufficient to partially rescue the *CaSR* loss-of-function mutant phenotype (Fig 5D). By contrast, CaSR expression driven by the negative control line did not rescue the *CaSR* mutant phenotype (Fig 5E). In summary, we used a series of partially overlapping Gal4 lines to show that CaSR activity specifically in the DCR6 is a key determinant of decision-making bias. This step demonstrates a causal role for CaSR in the brain region that was initially identified via the MAVEN strategy. Our example illustrates the utility of this strategy to identify Gal4 drivers for genetically-identifiable neuronal populations that mediate genetic effects on behavior.

## Optional step: Identifying other candidate regions

We validated the functional significance first candidate region that MAVEN identified. However, since the MAVEN strategy identifies candidates via correlation, this may not always be the case for users of this protocol. If validation in the first candidate region fails, the next steps depend on the results of the LASSO regression. For our analysis, LASSO regression returned only a single hit, but it is possible that in other cases the analysis will assign coefficients to multiple regions. If this occurs, and if driving expression only in the first candidate region fails to reproduce the phenotype of interest, either a less specific Gal4 or two Gal4s in combination could be used to drive expression in the top two candidate regions. This may reveal a requirement for a gene in multiple brain regions simultaneously.

If all candidate regions from the LASSO regression have been tested without successful rescue, there are still options available. LASSO regression tends to produce sparse models, which means that if multiple brain regions are highly correlated with each other and with the phenotype, some may be discarded from the model. Thus, multiple highly phenotype-linked candidate regions may be obscured by a single highest-linked candidate [25]. One option is simply re-running the LASSO regression with the previous top candidate brain region's data omitted. Another option is employing our function, exportStrongestCorrelatedRegions, which identifies the regions that are most highly correlated with a given region (protocols.io Step 18.3). This output of this function can be input to our CustomBrainRegionStack function in Matlab (protocols.io Step 18.6) to visualize the brain regions where expression is most highly correlated with the top LASSO hit region (Fig 6A). When we performed this analysis, after excluding brain regions that physically included the DCR6 (Rhombomere 6 and Rhombencephalon),

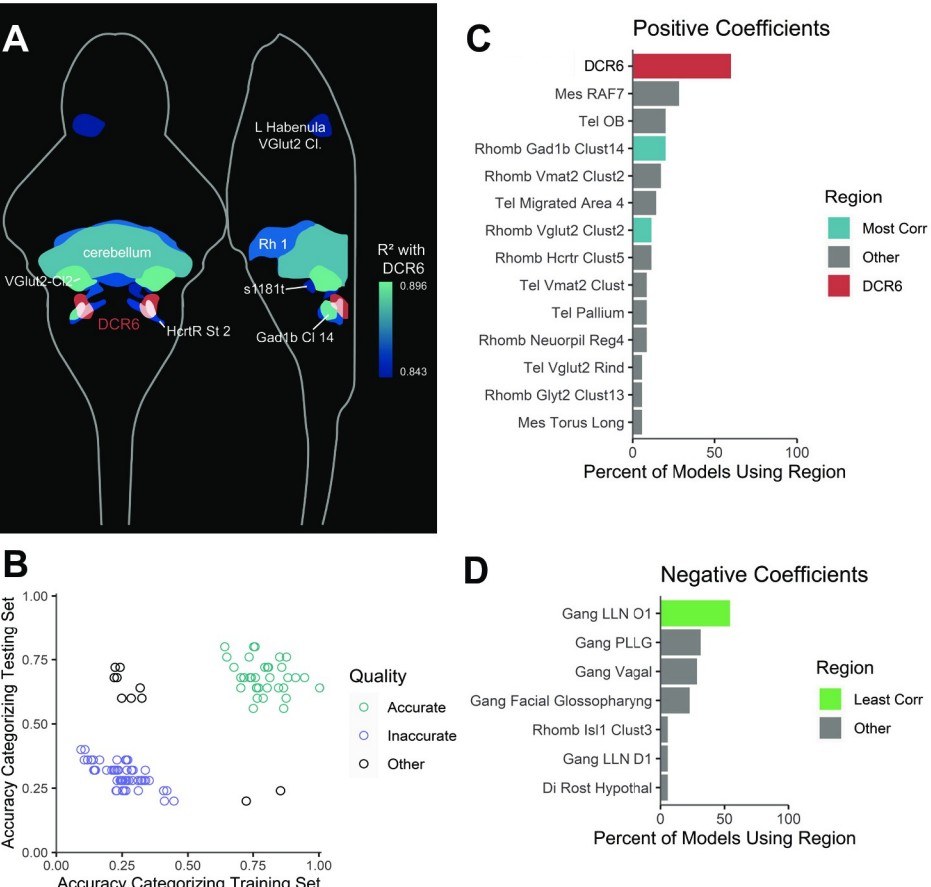

**Fig 6. Methods for identifying alternative candidate regions.** A) Anatomical map of regions with the highest correlations (Pearson $R^2$) with the DCR6. The DCR6 is indicated in red; correlating regions are indicated in shades of blue, with brighter color indicating greater correlation. The majority of regions that correlate highly with the DCR6 are nearby within the dorsal rhombencephalon. Some regions were excluded due to anatomical overlap with regions shown or because they physically contain the entire DCR6; see Supplemental Information full results. Abbreviations, in descending order of correlation strength: DCR6 –Dorsal cluster Rhombomere 6 ("Rhombencephalon—QRFP Neuron Cluster Sparse"); VGlut2 Cl2 –"Rhombencephalon—VGlut2 Cluster 2"; Gad1b Cl 14 –"Rhombencephalon—Gad1b Cluster 14"; Cerebellum–"Rhombencephalon—Cerebellum"; Rh 1 –"Rhombencephalon—Rhombomere 1"; HcrtR St 2 –"Rhombencephalon—6.7 DHCrtR Gal4 Stripe 2"; s1181t –"Rhombencephalon—s1181t Cluster"; L Habenula VGlut2 Cl–"Diencephalon—Left habenula VGlut2 Cluster" B) Training and testing set accuracy of 100 LASSO regression models created by bootstrapping data from n = 50 CaSR WT larvae expressing UAS:CaSR-EGFP under control of αtubulin:Gal4. Most models fall into one of two categories: accurate (greater than 50% accuracy on both training and testing sets, sea green, n = 35) or inaccurate (less than 50% accuracy on both training and testing sets, purple, n = 55). Datapoints are jittered to avoid overplotting. We hypothesize that the inaccurate models are the result of declining statistical power associated with halving sample size for bootstrapping. For all further analysis, only accurate models were used. C) Regions that received positive coefficients in more than one bootstrapped, accurate LASSO model. The most common region employed in accurate models was the DCR6 (red). Two other regions that are among the top 10 most correlated with the DCR6 (see Fig 5A and S1 File) also appeared (blue). The other regions (gray) may explain some of the variability in phenotype that was not explained by the DCR6. Abbreviations: DCR6 –"Dorsal Cluster Rhombomere 6" (ZBrain: "Rhombencephalon–QRFP Neuronal Cluster Sparse"); Mes RAF7 –"Mesencephalon—Retinal Arborization Field 7 (AF7)"; Tel OB–"Telencephalon—Olfactory Bulb"; Rhomb Gad1b Clust14 –"Rhombencephalon—Gad1b Cluster 14"; Rhomb Vmat2 Clust2 –"Rhombencephalon—Vmat2 Cluster 2"; Tel Migrated Area 4 –"Telencephalon—Telencephalic Migrated Area 4 (M4)"; Rhomb Vglut2 Clust2 –"Rhombencephalon—Vglut2 cluster 2; Rhomb Hcrtr Clust5 –"Rhombencephalon—6.7FDhcrtR-Gal4 Cluster 5"; Tel Vmat2 Clust–"Telencephalon—Vmat2 Cluster"; Tel Pallium–"Telencephalon—Vmat2 Cluster"; Rhomb Neuropil Reg4 –"Rhombencephalon—Neuropil Region 4"; Tel Vglut2 Rind–"Telencephalon—Vglut2 rind"; Rhomb Glyt2 Clust13 –"Rhombencephalon—Glyt2 Cluster 13"; Mes Torus Long–"Mesencephalon—Torus Longitudinalis". D) Regions that received negative coefficients in more than one bootstrapped, accurate LASSO model. The most commonly employed region was lateral line neuromast O1, which is the single least correlated region with the DCR6 (green, top 10 regions least correlated with DCR6). Other regions are shown in gray. Abbreviations: Gang LLN O1

–"Ganglia—Lateral Line Neuromast O1"; Gang PLLG–"Ganglia—Posterior Lateral Line Ganglia"; Gang Vagal–"Ganglia—Vagal Ganglia"; Gang Facial Glossopharyng–"Ganglia—Facial glossopharyngeal ganglion"; Rhomb Isl1 Clust3 –"Rhombencephalon—Isl1 Cluster 3"; Gang LLN D1 –"Ganglia—Lateral Line Neuromast D1"; Di Rost Hypothal–"Diencephalon—Rostral Hypothalamus".

we were left with predominantly dorsal hindbrain regions, including other nuclei within Rhombomere 6 and the cerebellum (S1 File).

An estimate for reliability of a LASSO regression model can be obtained by bootstrapping the data. When we bootstrapped our LASSO regression, we used half of our original dataset (n = 25 *CaSR* WT brains) for training and half for testing and created 100 models from the randomly subsampled data. The resulting models exhibited a surprising bimodal distribution of accuracy. Some models (n = 35/100) accurately classified both the training and testing dataset, while others (n = 55/100) failed to accurately classify either (Fig 6B). We speculate that halving our dataset for the bootstrapping caused such a great loss in statistical power that it was no longer possible to produce an accurate model from the data in most cases. Therefore, for all subsequent analyses, we use only the bootstrapped models that produced accurate results (n = 35 models).

Our original analysis on the full dataset highlighted a single brain region, the DCR6. 60% (n = 21/35) of our accurate bootstrapped models also incorporated the DCR6, making it the single region most likely to be employed by an accurate model. We would expect that if the DCR6 was the only region where signal predicts phenotype, our bootstrapped models would tend to employ either the DCR6 or regions very highly correlated with the DCR6. Bootstrapping, however, revealed other regions where CaSR-EGFP signal could potentially be used to predict phenotype (Fig 6C), only two of which were among the 10 brain regions most highly correlated with the DCR6. For those seeking to generate alternative candidate regions if the initial LASSO regression fails to validate, bootstrapping may reveal hidden possibilities.

When we bootstrapped our analysis, we also found a few regions that our accurate models used as negative coefficients (e.g. more signal in the region predicts the typical phenotype, not the overexpression phenotype). For example, 54% (n = 19/35) of our accurate models assigned a negative coefficient to lateral line neuromast O1, the single region least correlated with the DCR6 (Fig 6D). Overall, our results suggest that phenotypic regulation, even by a single gene, may be highly spatially complex. While overexpression of a gene in the whole brain may shift a phenotype in one direction, overexpression in specific regions may have the opposite effect. Therefore, it is not surprising that both positive and negative coefficients may be employed in a model for phenotype prediction.

## Strengths and limitations of the MAVEN strategy

We selected LASSO regression to detect potentially important brain regions from a large set of possible regions. However, LASSO regression has its own strengths and limitations relative to other forms of multivariate analysis. Because LASSO regression produces an additive model, it may be able to detect effects of gene's expression levels in multiple regions, especially if the analysis is highly powered. This is a potential strength of the analysis, as it allows for the possible identification of multiple brain regions that may only produce phenotypic effects in combination with each other, a capability that simple Gal4 x UAS screens do not have. However, more complex scenarios may not be best detected by LASSO regression. For example, if gene expression in any one of a several brain regions is sufficient to produce a

phenotype, LASSO regression may not be able to construct a suitable model. Other types of models may prove more suitable in such cases. In this case, variegated expression data could be generated and quantified using the methods we describe in this paper, but an alternative multivariate model, such as a neural network or a random forest model, could be employed at the analysis stage.

Another limitation arises from using anatomically defined brain regions as the basic unit of the analysis. Within each brain region, there are multiple cell types with differing neurotransmitter identities, patterns of connectivity, and functional activity. If only a small fraction of the cells in a given brain region functionally contribute to a given genetic phenotype, the MAVEN strategy may fail to identify that region as important. Relatedly, if a relevant population of cells spans multiple brain regions, MAVEN may not identify both regions. Therefore, failure to be identified by the MAVEN strategy should never be considered evidence that a brain region does not contribute to a genetic phenotype, particularly if that region is functionally heterogeneous. Along the same line of logic, when validating regions identified with specific Gal4 x UAS experiments, we recommend attempting validation with multiple different Gal4s, as they may express at different strengths, in different subsets of neurons within the region in question, and/or in different sets of off-target regions.

## Future improvement of the MAVEN strategy

One of the requirements to attempt MAVEN is a variegating or mosaic expression construct for the gene of interest. While we were able to achieve this serendipitously due to a variegated UAS construct, we realize that this may not always be available. Fortunately, other methods are available for deliberately inducing mosaic gene expression in zebrafish. Microinjecting genetic constructs into zebrafish embryos very early in development (1–4 cell stage) results in mosaic animals, and use of cell-type-specific promoters enables largely cell-type-specific mosaic expression [27–29]. Injection of a labeled rescue construct thus is one potential avenue for generating cell-type and/or tissue-specific mosaic expression.

Other methods for mosaic expression make use of stochastic activity of recombinases. For example, in zebrafish *UAS:Switch*, which recombines Cre-dependently [30], and *UAS:bloSwitch*, which recombines B3-dependently [31], have both been successfully used to label neurons within a genetically-defined population [31]. In both cases, a GFP-STOP element is flanked by recombination sites. When the GFP-STOP element is recombined out, a downstream RFP element can then be transcribed. By adding an expression construct for the gene of interest prior to the RFP sequence, either a Switch or bloSwitch construct could be repurposed for stochastic expression of a rescue construct in a genetically-defined subpopulation of cells. Furthermore, the dosage of a heat shock-inducible or injected recombinase could be titrated to ensure roughly equal-sized groups of rescued and non-rescued larvae for highest experimental efficiency. Other transgenic techniques for generating mosaic zebrafish, including MAZe [32], zMADM [33], and Bi-FoRe [34], may also prove adaptable for MAVEN analysis.

A major aim of neuroscience is to understand the cellular and circuit mechanisms by which specific genetic mutations affect behavioral and neurological phenotypes. Fulfilling this aim requires determination of the specific neurons in which a gene's function is relevant to behavior. Previously, accomplishing this task in larval zebrafish has required arduous, time-consuming cell-type-specific rescue screens. We have developed an unbiased strategy, MAVEN, for prioritizing brain regions in which rescue experiments are most likely to succeed, then validating causal roles for these regions in behavior. Our MAVEN strategy thereby facilitates the integration of molecular-genetic, circuit, and behavioral neurobiology.

## Supporting information

**S1 Protocol. Protocols.io DOI. dx.doi.org/10.17504/protocols.io.14egn78jyv5d/v1.**
(PDF)

**S1 File. Data used for figure creation.**
(XLSX)

**S1 Code. R and Matlab code and associated files (also available at protocols.io).**
(ZIP)

## Acknowledgments

We gratefully acknowledge Dr. Eugenio Piasini of the UPenn Computational Neuroscience Initiative for providing guidance on computational strategies, Dr. Daniel Beiting and his Spring 2020 DIY Transcriptomics class for providing an excellent resource for learning R, Dr. Andrea Stout at the UPenn CDB Microscopy Core (RRID:SCR_022373) and Dr. Joseph Zinski for microscopy advice, the UPenn fish facility staff (especially for their continuous work during the pandemic), Elelbin Ortiz for protocol testing, and Dr. Mike Iannacone, Dr. Jessica Nelson, and Dr. Joy Meserve for their helpful comments on a draft of this manuscript.

## Author Contributions

**Conceptualization:** Hannah Shoenhard.

**Formal analysis:** Hannah Shoenhard.

**Funding acquisition:** Michael Granato.

**Investigation:** Hannah Shoenhard.

**Methodology:** Hannah Shoenhard.

**Project administration:** Michael Granato.

**Software:** Hannah Shoenhard.

**Supervision:** Michael Granato.

**Validation:** Hannah Shoenhard.

**Visualization:** Hannah Shoenhard.

**Writing – original draft:** Hannah Shoenhard.

**Writing – review & editing:** Hannah Shoenhard, Michael Granato.

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
