## [Decision Letter · Decision Letter 0]

16 Jan 2023

PONE-D-22-30720

Multivariate Analysis of Variegated Expression in Neurons: a strategy for unbiased localization of gene function to candidate brain regions in larval zebrafish

PLOS ONE

Dear Dr. Granato,

Thank you for submitting your manuscript to PLOS ONE. After careful consideration, we feel that it has merit but does not fully meet PLOS ONE’s publication criteria as it currently stands. Therefore, we invite you to submit a revised version of the manuscript that addresses the points raised during the review process.

We look forward to receiving your revised manuscript.

Kind regards,

Bruno Giros, Ph.D.

Academic Editor

PLOS ONE

Journal Requirements:

Reviewers' comments:

Reviewer's Responses to Questions

**Comments to the Author**

1. Does the manuscript report a protocol which is of utility to the research community and adds value to the published literature?

Reviewer #1: Yes

 2. Has the protocol been described in sufficient detail?

To answer this question, please click the link to protocols.io in the Materials and Methods section of the manuscript (if a link has been provided) or consult the step-by-step protocol in the Supporting Information files.

The step-by-step protocol should contain sufficient detail for another researcher to be able to reproduce all experiments and analyses.

Reviewer #1: Yes

3. Does the protocol describe a validated method?

Reviewer #1: Yes

 4. If the manuscript contains new data, have the authors made this data fully available?

Reviewer #1: Yes

 **5. Is the article presented in an intelligible fashion and written in standard English?**

Reviewer #1: Yes

6. Review Comments to the Author

Reviewer #1: This work describes the use of a protocol developed by the authors to identify a zebrafish brain region in which the CaSR gene acts to control response to an acoustic stimulus (reorientation versus escape). The technique, MAVEN, is an unbiased correlative approach that relies on variability of expression of Gal4 driver lines in zebrafish brains. In other words, each transgenic larva expresses the driver in a different set of brain regions. Sorting animals by behavioral phenotype and then correlating regions of brain expression is the basic idea, but a thorough set of statistical and computational analyses is employed to derive a correlation of expression to behavioral phenotype.

This work is very important because, especially in vertebrate behavior, connecting phenotype to brain region is, as the authors say, a bottleneck. This technique turns a disadvantage (variegated transgene expression) into an advantage to make these phenotype-expression region correlations that can relatively quickly identify the brain region.

The work identified the DCR6 (rhombomere 6) region of the brain as correlating with rescue. Known DCR6 Gal4 expression lines were then directly used to confirm the importance of CaSR expression in that region.

This work should definitely be published so that the greater community can begin to use it to determine its broad utility. The protocol has already been published on protocols.oi.

Comment:

The “gain-of-function” and “loss-of-function” approaches could be better explained when first mentioned. “Loss-of-function”, I assumed, means rescue of a mutant, but it was not stated exactly what “gain-of-function” means (e.g. overexpression?). There seems to be some intermingling of these concepts when the results are described.

7. PLOS authors have the option to publish the peer review history of their article (what does this mean?). If published, this will include your full peer review and any attached files.

Reviewer #1: No

---

## [Author Response · Author response to Decision Letter 0]

19 Jan 2023

We appreciate the reviewer for their time and thoughtful comments on our manuscript. The reviewer requests clarification regarding our use of the term “gain-of-function” phenotype as opposed to “overexpression phenotype.” We agree that our initial choice of language was potentially confusing. According to the reviewer’s suggestion, we have changed the language throughout the manuscript to reflect that we performed our analysis on larvae that overexpressed CaSR in neurons in a variegated fashion, and that this overexpression produced a behavioral phenotype. We now avoid the term “gain-of-function” to avoid potential confusion with gain-of-function mutations, which we did not employ in this manuscript. We thank the reviewer for their helpful suggestion.

---

## [Editor Report · Decision Letter 1]

27 Jan 2023

Multivariate Analysis of Variegated Expression in Neurons: a strategy for unbiased localization of gene function to candidate brain regions in larval zebrafish

PONE-D-22-30720R1

Dear Dr. Granato,

We’re pleased to inform you that your manuscript has been judged scientifically suitable for publication and will be formally accepted for publication once it meets all outstanding technical requirements.

Kind regards,

Bruno Giros, Ph.D.

McGill University

Academic Editor

PLOS ONE
---

## [Editor Report · Acceptance letter]

3 Feb 2023

PONE-D-22-30720R1 

Multivariate Analysis of Variegated Expression in Neurons: a strategy for unbiased localization of gene function to candidate brain regions in larval zebrafish 

Dear Dr. Granato:

I'm pleased to inform you that your manuscript has been deemed suitable for publication in PLOS ONE. Congratulations! Your manuscript is now with our production department. 

Kind regards, 

on behalf of

Professor Bruno Giros 

Academic Editor

PLOS ONE